# Genotyping of *Coxiella burnetii* from Cattle by Multispacer Sequence Typing and Multiple Locus Variable Number of Tandem Repeat Analysis in the Republic of Korea

**DOI:** 10.3390/genes13111927

**Published:** 2022-10-23

**Authors:** A-Tai Truong, So Youn Youn, Mi-Sun Yoo, Ji-Yeon Lim, Soon-Seek Yoon, Yun Sang Cho

**Affiliations:** 1Parasitic and Honeybee Disease Laboratory, Bacterial Disease Division, Department of Animal & Plant Health Research, Animal and Plant Quarantine Agency, 177 Hyeoksin 8-ro, Gimcheon-si 39660, Gyeongsangbuk-do, Korea; 2Faculty of Biotechnology, Thai Nguyen University of Sciences, Thai Nguyen 250000, Vietnam

**Keywords:** *Coxiella burnetii*, multiple locus variable number tandem repeat analysis, multispacer sequence typing, Q fever

## Abstract

Genotyping of *Coxiella burnetii* using multispacer sequence typing (MST) and multiple locus variable number tandem repeat analysis (MLVA) was conducted from infected animals for the first time in the Republic of Korea. *C. burnetii* was detected by real-time PCR, and followed by MST and MLVA genotyping. The result showed that detected *C. burnetii* all had the same MLVA genotype, 6-13-2-7-9-10 for markers MS23-MS24-MS27-MS28-MS33-MS34, respectively, and genotype group 61 for MST. The same genotypes were previously identified in Poland. Importantly, this MLVA type was detected in humans in France, suggesting that the Korean strain can also potentially cause Q fever in humans. MST and MLVA were very useful tools for analyzing the molecular epidemiology of *C. burnetii* and helpful for interpreting the epidemiological relationship between isolates from domestic and international resources.

## 1. Introduction

*C. burnetii* is an obligate intracellular bacterium, a causative agent of Query (Q) fever in human and animals [1]. *C. burnetii*-infected people might have clinical illness after one to three weeks, with symptoms of flu, fever, headache, diarrhea, night sweats, and abdominal or chest pains [2]. The zoonotic disease has been dispersed worldwide and has become an emerging threat to public health [3]. Ruminants are the main natural reservoirs of *C. burnetii*, which can be found in urine, faeces, milk, or birth products of infected animals [1,4,5]. Human infection mainly occurs via inhalation of contaminated aerosols. However, consumption of raw animal product can also result in the infection of *C. burnetii* [6]. Identification of the sources of *C. burnetii* and molecular characterization of the bacterial strain is necessary to identify the link between epidemiological sources and human infection [7,8].

Several techniques have been developed for genotyping and characterization of *C. burnetii*. Earlier developed methods were based on the analysis of PCR-restriction fragment length polymorphism [9,10], restriction-endonuclease-digested DNA separated by SDS-PAGE [11], and sequence analysis of the *com1* and *mucZ* genes [12]. However, these methods have a limitation of inter- and intra-laboratory reproducibility [13]. Afterwards, the high-resolution multispacer sequence typing (MST) [14] and multiple locus variable number tandem repeat analysis (MLVA) methods [15,16] were developed and used extensively to date for epidemiological tracking and for identifying the phylogenetic relationship of *C. burnetii* isolates from different regions [17,18].

In the Republic of Korea (ROK), *C. burnetii* has been widely detected in cattle [19], and in ticks infesting cattle [20]. *C. burnetii* infection in a human who had contact with cattle was reported in 2020 [21]. The patient worked as a dairy cattle raiser and was diagnosed with *C. burnetii* infection after one month of fever. Therefore, the pathogen in infected cattle could be a potential source of human infection. However, a molecular method to identify the circulation of the bacterium in infected animals and humans has not been introduced in the country, and therefore it is necessary to determine the sources of human infection and the relationship between *C. burnetii* isolates from domestic and international resources. Therefore, this study was conducted to characterize the genotype of *C. burnetii* detected in vaginal swabs of cattle in ROK using MST and MLVA.

## 2. Materials and Methods

### 2.1. Samples

A total of eight vaginal swab samples from cattle with *C. burnetii* infection were used for genotyping analysis. The samples were collected from cattle in the Jeonnam (*n* = 1) and Chungnam (*n* = 7) provinces in ROK in 2021.

### 2.2. DNA Extraction and Detection of C. burnetii

DNA extraction from swab samples was performed using QIAamp Fast DNA Stool Mini Kit (QIAGEN, Hilden, Germany). The procedure of DNA extraction was performed according to the instructions from the kit. Detection of *C. burnetii* in the samples was performed by ultra-rapid real-time PCR targeting on IS1111 insertion element using primer Cox-F (5′-GTCTTAAGGTGGGCTGCGTG-3′), and Cox-R (5′-CCCCGAATCTCATTGATCAGC-3′); and probe Cox-TM (FAM-AGCGAACCATTGGTATCGGACGTT-TAMRA) [20,22].

### 2.3. Multiple Locus Variable Number Tandem Repeat Analysis (MLVA)

MLVA was performed by analyzing six microsatellite markers (MS23, MS24, MS27, MS28, MS33, and MS34) [15]. Primers used for amplification of the markers are shown in Table 1. PCR was performed using AccuPower^®^ ProFi *Taq* PCR PreMix (Bioneer, Daejeon, Korea), and the 20 µL reaction mix consisted of 1 µL (10 pmol) of each primer, 3 µL DNA template, and 15 µL of ddH_2_O. PCR conditions were 95 °C (3 min), 37 cycles of 95 °C (30 s)–57 °C (30 s)–72 °C (30 s), and 72 °C (5 min). The PCR product of each marker from different samples was loaded in 4% (*w*/*v*) agarose gel for electrophoresis to identify the different amplicon size of each marker among the samples. The band in agarose gel was extracted for sequencing analysis. The MS23, MS24, MS28, and MS34 amplicons were directly sequenced using the specific primers; meanwhile, the amplicons of MS27 and MS33 were inserted into plasmid pGEM-T using the pGEM-T^®^ easy vector system (Promega, Madison, WI, USA). The recombinant plasmids carrying MS27 and MS33 were introduced into *Escherichia coli* DH5α (Enzynomics, Daejeon, Korea) and grown in Luria-Bertani medium. The recombinant plasmids were extracted using an AccuPrep^®^ plasmid mini extraction kit (Bioneer, Daejeon, Korea) and were sequenced using primer M13F (5′-GTA AAA CGA CGG CCA GTG-3′) and M13R (5′-CAG GAA ACA GCT ATG AC-3′) due to the small size of the two markers. Sanger sequencing was performed by Macrogen Inc. (Seoul, Korea). The generated sequence data was aligned with reference sequences of the Nine Mile strain using Clustal X 2.0 [23] to identify the number of tandem repeats. The genotype of the Nine Mile strain is 9-27-4-6-9-5 for markers MS23-MS24-MS27-MS28-MS33-MS34. The genotype was identified by comparing to the MLVA database, available at https://microbesgenotyping.i2bc.paris-saclay.fr/databases (accessed on 20 August 2022).

### 2.4. Multispacer Sequence Typing (MST)

Ten spacers used for analyzing of MST were previously described by Glazunova et al. [14]. The spacers include Cox2, Cox5, Cox18, Cox20, Cox22, Cox37, Cox51, Cox56, Cox57, and Cox61. Primers used for amplification and sequencing of the ten spacers are shown in Table 2. PCR was performed using AccuPower^®^ ProFi *Taq* PCR PreMix (Bioneer, Daejeon, Korea), and the 20 µL reaction mix was composed of 1 µL (10 pmol) of each primer, 3 µL DNA template, and 15 µL of ddH_2_O. PCR conditions were 95 °C (3 min), 37 cycles of 95 °C (30 s)–55 °C (30 s)–72 °C (1 min), and 72 °C (5 min). PCR products were sequenced by Macrogen Inc. (Seoul, Korea). The genotype was identified by comparing to the MST database available at https://ifr48.timone.univ-mrs.fr/mst/coxiella_burnetii/blast.html (accessed on 20 August 2022). Phylogenetic tree showing the relationship among the genotypes was created using MEGA7 software [24].

## 3. Results

The result of electrophoresis showed that *C. burnetii* in all eight samples collected from the Jeonnam and Chungnam provinces had the same MLVA type with the same amplicon size of six markers (Figure 1). Sequencing analysis of the six markers from a representative sample, D25 (Figure 2), showed that the genotype of Korean *C. burnetii* strains is 6-13-2-7-9-10 for markers MS23-MS24-MS27-MS28-MS33-MS34, respectively. This genotype was also reported in various countries in Europe such as the Netherlands, France, Spain, and Poland (Figure 3).

The 10 spacers of MST were amplified and confirmed in agarose gel electrophoresis (Figure 4). Sequencing analysis of the spacers from eight samples showed that *C. burnetii* in all samples had the same genotype, 3-2-6-1-5-10-4-10-6-5, for the spacers Cox2-Cox5-Cox18-Cox20-Cox22-Cox37-Cox51-Cox56-Cox57-Cox61, respectively. The genotype of the detected *C. burnetii* belongs to MST group 61, which was reported in cattle milk from Poland and Iran. The genotype has a close relationship with the genotype of MST group 20, which originated from France, Germany, USA, and Spain (Figure 5). The combination of MST and MLVA showed that the strain of *C. burnetii* identified in South Korea had a close relationship with the strain Cb_PL06 identified in Poland in 2015 with the same MST (MST61) and MLVA type (Figure 3).

## 4. Discussion

MST and MLVA genotyping of *C. burnetii* was performed in this study for the first time in South Korea. Only one genotype of *C. burnetii* was seen from samples collected in two provinces, Jeonnam and Chungnam. The genotype of *C. burnetii* detected in this study had the same type of MST and MLVA as that detected in cattle milk in Poland [25]. The result suggests that the pathogen in ROK could have originated from Poland.

MST has been established for identification of the geographical relationship of *C. burnetii*. Meanwhile, MLVA is useful for epidemiological purposes [8,26]. The MLVA genotype identified in this study was detected in humans in France in 2000 (https://microbesgenotyping.i2bc.paris-saclay.fr/databases/view/43 (accessed on 20 August 2022)). The result suggests that the detected *C. burnetii* strain in cattle in the two provinces could be an important source of causative agents of Q fever in humans in ROK. In addition, *C. burnetii* was detected and isolated from a patient who worked as a dairy cattle raiser in ROK [21,27]. However, the evidence of direct transmission of *C. burnetii* from cattle has not been provided. MST and MLVA typing could be useful to determine the relationship between *C. burnetii* in its vectors, natural reservoirs, and in patients. Therefore, the result of this study initially provides important information for further study on the epidemiological source of Q fever in ROK.

MST and MLVA genotyping of *C. burnetii* in ROK provide information on geographical distributions of the *C. burnetii* genotypes, and it is important to establish the global database on the phylogenetic relationship among the geographical strains and epidemiological study on Q fever. Furthermore, it could be useful information for determining the relationship between human infection and the natural reservoirs of the pathogens, by which a high potential source of disease could be suggested.

## 5. Conclusions

The genotype of *C. burnetii* based on MST and MLVA was analyzed in ROK for the first time. The group MST61 was identified with the code 3-2-6-1-5-10-4-10-6-5 for spacers Cox2-Cox5-Cox18-Cox20-Cox22-Cox37-Cox51-Cox56-Cox57-Cox61, respectively. In addition, MLVA typing was identified with the tandem repeat 6-13-2-7-9-10 for markers MS23-MS24-MS27-MS28-MS33-MS34, respectively. The same MST and MLVA type of *C. burnetii* from cattle in Poland was seen. The result demonstrated a close relationship of *C. burnetii* strains between the two countries. Furthermore, a close relationship between MST61 and MST20, which includes strains that have infected humans, from the same type of MLVA, suggests that the strain identified in ROK could potentially transmit to humans. This study provides the initial result of MST and MLVA typing of *C. burnetii* that is helpful for analyzing molecular epidemiology of *C. burnetii* in the country.

## Figures and Tables

**Figure 1 genes-13-01927-f001:**
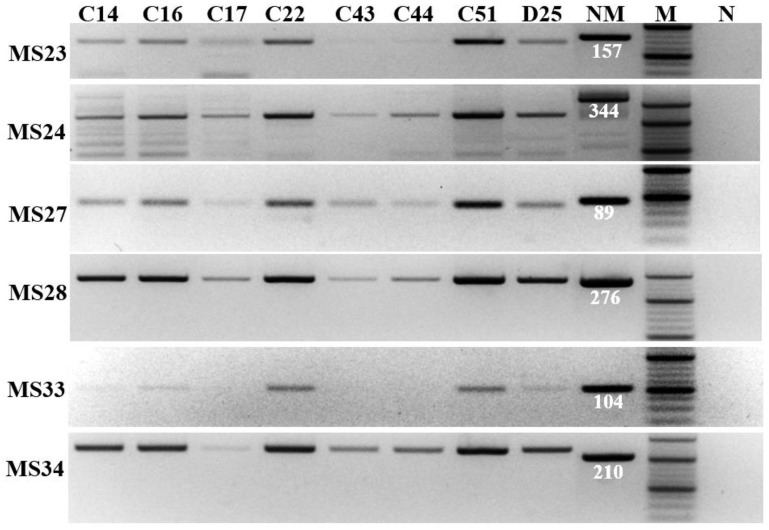
MLVA typing of *C. burnetii*. PCR products of six markers (MS23, MS24, MS27, MS28, MS33, and MS34) were amplified from eight vaginal swab samples: C14, C16, C17, C22, C43, C44, C51, and D25. The reference Nine Mile strain (NM) with amplicon size (in base pair long) of each marker is indicated, “M” is 20 bp DNA marker, and “N” is negative control without DNA template.

**Figure 2 genes-13-01927-f002:**
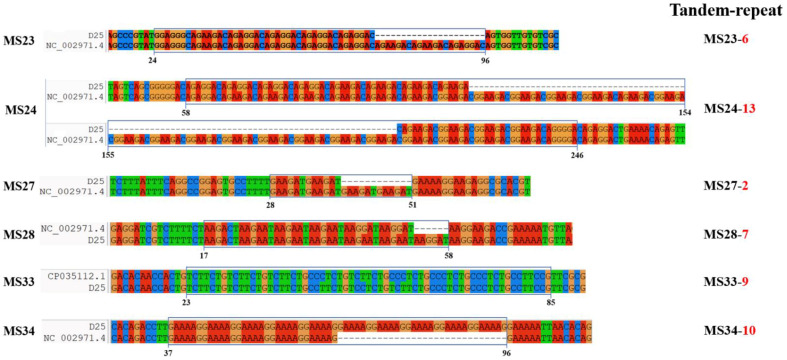
Identification of tandem repeat of *C. burnetii*. Alignment of sequences of six markers (MS23, MS24, MS27, MS28, MS33, and MS34) from the Nile Mile strain (NCBI accession No.: NC_002971 and CP035112) and the detected strain D25 is shown. The region with repeat units in each marker is indicated, and the numbers indicating the positions in the analyzed sequences are presented. Number of tandem repeats of each marker is shown.

**Figure 3 genes-13-01927-f003:**
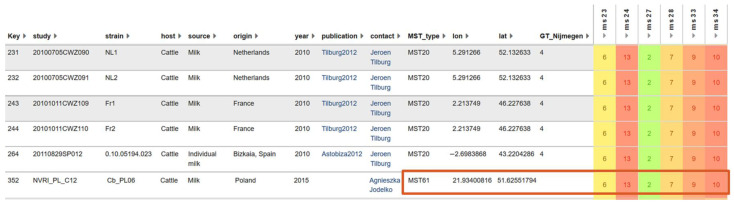
Identification of MLVA type of *C. burnetii* detected in South Korea. Six markers (MS23, MS24, MS27, MS28, MS33, and MS34) of MLVA of *C. burnetii* identified in South Korea were compared with the MLVA database. Strains identified in the Netherlands, France, Spain, and Poland showed the same MVLA type, of which the strain Cb_PL06 had the same MST (MST61) and MLVA type with the strain identified in South Korea (marked in the red box).

**Figure 4 genes-13-01927-f004:**
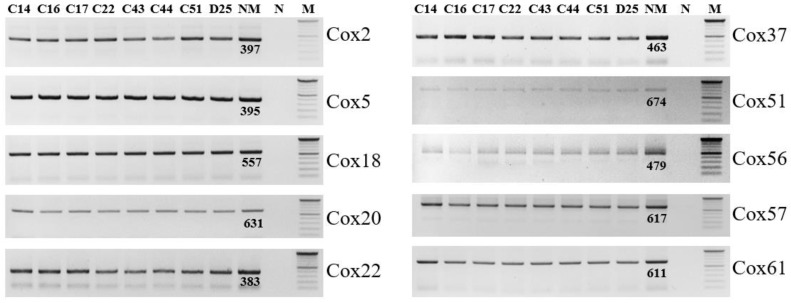
Amplification of 10 spacers for MST analysis. The 10 spacers, including Cox2, Cox5, Cox18, Cox20, Cox22, Cox37, Cox51, Cox56, Cox57, and Cox61, were amplified from eight vaginal swab samples: C14, C16, C17, C22, C43, C44, C51, and D25. “NM” is the Nine Mile strain with amplicon size (in base pair long) of each marker is shown, “M” is 100 bp DNA marker, and “N” is negative control without DNA template.

**Figure 5 genes-13-01927-f005:**
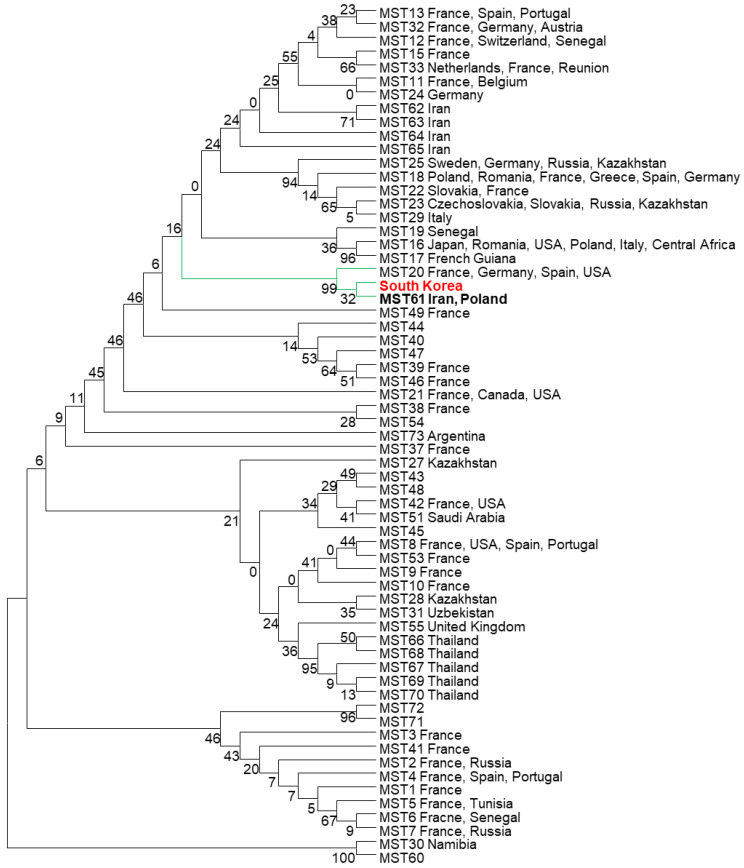
Phylogenetic tree showing relationships among the different groups of MST. Maximum-likelihood phylogenetic trees were created from sequences of *C. burnetii* detected from vaginal swab samples of cattle in South Korea and other MST groups using the Kimura 2-parameter model, γ distribution, and bootstrapping 500 times with MEGA7 software.

**Table 1 genes-13-01927-t001:** Primers used for multiple locus variable-number tandem-repeat analysis (MLVA).

Locus	Primer	Sequence (5′–3′)	Amplicon Size (bp)	Reference
Ms23	Ms33-F	GGACAAAAATCAATAGCCCGTA	157	[8,15]
	Ms33-R	GAAAACAGAGTTGTGTGGCTTC		
Ms24	Ms24-F	ATGAAGAAAGGATGGAGGGACT	344	
	Ms24-R	GATAGCCTGGACAGAGGACAGT		
Ms27	Ms27-F	TCTTTATTTCAGGCCGGAGT	89	
	Ms27-R	GAACGACTCATTGAACACACG		
Ms28	Ms28-F	TAGCAAAGAAATGTGAGGATCG	276	
	Ms28-R	ATTGAGCGAGAGAATCCGAATA		
Ms33	Ms33-F	TCGCGTAGCGACACAACC	104	
	Ms33-R	GTAGCCCGTATGACGCGAAC		
Ms34	Ms34-F	TGACTATCAGCGACTCGAAGAA	210	
	Ms34-R	TCGTGCGTTAGTGTGCTTATCT		

**Table 2 genes-13-01927-t002:** Primers for multispacer sequence typing (MST).

Spacer Name	Primer	Sequence (5′–3′)	Amplicon Size (bp)	Reference
Cox2	Cox20766	CAACCCTGAATACCCAAGGA	397	[14]
	Cox21004	GAAGCTTCTGATAGGCGGGA		
Cox5	Cox77554	CAGGAGCAAGCTTGAATGCG	395	
	Cox77808	TGGTATGACAACCCGTCATG		
Cox18	Cox283060	CGCAGACGAATTAGCCAATC	557	
	Cox283490	TTCGATGATCCGATGGCCTT		
Cox20	Cox365301	GATATTTATCAGCGTCAAAGCAA	631	
	Cox365803	TCTATTATTGCAATGCAAGTGG		
Cox22	Cox378718	GGGAATAAGAGAGTTAGCTCA	383	
	Cox378965	CGCAAATTTCGGCACAGACC		
Cox37	Cox657471	GGCTTGTCTGGTGTAACTGT	463	
	Cox657794	ATTCCGGGACCTTCGTTAAC		
Cox51	Cox824598	TAACGCCCGAGAGCTCAGAA	674	
	Cox825124	GCGAGAACCGAATTGCTATC		
Cox56	Cox886418	CCAAGCTCTCTGTGCCCAAT	479	
	Cox886784	ATGCGCCAGAAACGCATAGG		
Cox57	Cox892828	TGGAAATGGAAGGCGGATTC	617	
	Cox893316	GGTTGGAAGGCGTAAGCCTTT		
Cox61	Cox956825	GAAGATAGAGCGGCAAGGAT	611	
	Cox957249	GGGATTTCAACTTCCGATAGA		

## Data Availability

All data generated or analyzed during this study are included in this article.

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
