# Peer review of "Genotyping of Coxiella burnetii from Cattle by Multispacer Sequence Typing and Multiple Locus Variable Number of Tandem Repeat Analysis in the Republic of Korea"

_genes, 2022, doi:10.3390/genes13111927_

Round 1
Reviewer 1 Report
In this manuscript, the authors performed genotypic characterisation of Coxiella burnetii detected in the vaginal samples collected from eight cattle animals via multispacer sequence typing and multiple locus variable number of tandem repeat analysis. It is an important work, but why only partial results were displayed in this manuscript. Of eight C. burnetii-positive samples examined, the authors showed and described only one set of MST and MLVA profiles after Sanger Sequencing and sequence analysis. What happened to the rest? Please include the missing data and update the results section including the multiple sequence alignment (is RSA 331 really needed since the authors included only the NM strain as their positive control in this assay?) and phylogenetic tree. The electrophoresis gel image of the MST amplicons for all samples as well as an additional table showing individual identified MST profile should be included. It is only with the above crucial amendments that this manuscript can be further considered for publication in Genes. Please see additional minor comments only up to the methods section at this stage as below.
L28: faeces
L34: The methods > Earlier developed methods were
L37-40: Afterwards, the high-resolution multispacer sequence typing (MST) [13] and multiple locus variable number tandem repeat analysis (MLVA) methods [14,15] were developed and used extensively to date for epidemiological tracking and for identifying the phylogenetic relationship of C. burnetii isolates from different regions.
L61: delete “of panel two”
L62: …, and the 20 µl reaction mix consisted of … and 15 µl of ddH2O.
L68: The MS23 … and MS34 amplicons were directly …
L69-71: Please describe how the MS27 and MS33 PCR products were ligated into pGEM-T. Also, please include M13F and M13R primer sequences.
L72: method was used > was performed
L72: The generated sequence data was aligned with reference … using Clustal X 2.0 to identify the number of tandem-repeat.
L74: The genotypes
L75: strains
L83: delete “with specific primer for PCR amplification and sequencing”
L86-87: …, and the 20 µl reaction mix was composed of … and 15 µl of ddH2O.
Reviewer 2 Report
The manuscript has been written quite well. However, to enhance the quality of the manuscript, authors may re-consider these comments:
1. To add abbreviation of C. burnetii at Line 1 after Coxiella burnetii
2. What do you mean by prophylaxis in Line 30?
3. To add significance of zoonotic implications of C.burnetii in Introduction
4. To add more information on human infection (e.g. year, history) to Line 42
5. Elaborate more for Line 43-44
6. How did these 8 infected cattle diagnosed with Q fever?
7. Any animal ethic clearance upon sampling from these animals?
8. Reference on one important study in Poland is not cited in the manuscript
Round 2
Reviewer 1 Report
Thank you to the authors who have addressed my earlier comments as appropriate. Without further question, I now endorse the publication of this manuscript in Genes.